# The Advantages of Radiofrequency Echographic MultiSpectrometry in the Evaluation of Bone Mineral Density in a Population with Osteoarthritis at the Lumbar Spine

**DOI:** 10.3390/diagnostics14050523

**Published:** 2024-03-01

**Authors:** Carla Caffarelli, Antonella Al Refaie, Caterina Mondillo, Giuditta Manasse, Alessandro Versienti, Maria Dea Tomai Pitinca, Edoardo Conticini, Bruno Frediani, Stefano Gonnelli

**Affiliations:** 1Section of Internal Medicine, Department of Medicine, Surgery and Neuroscience, University of Siena, 53100 Siena, Italy; antonella.alrefaie@student.unisi.it (A.A.R.); gonnelli@unisi.it (S.G.); 2Rheumatology Unit, Department of Medicine, Surgery and Neuroscience, University of Siena, 53100 Siena, Italy; conticini@student.unisi.it (E.C.); bruno.frediani@unisi.it (B.F.)

**Keywords:** osteoarthritis, bone mineral density (BMD), dual-energy X-ray absorptiometry (DXA), radiofrequency echographic multispectrometry (REMS), osteoporosis, Kellgren/Lawrence grading score

## Abstract

Background: Osteoarthritis (OA) in the lumbar spine can potentially lead to an overestimation of bone mineral density (BMD), and this can be a challenge in accurately diagnosing conditions like osteoporosis, where precise measurement of BMD is crucial. Radiofrequency Echographic Multi Spectrometry (REMS) is being recognized as an innovative diagnostic tool for assessing bone status. The purpose of this study was to evaluate whether the use of REMS may enhance the identification of osteoporosis in patients with osteoarthritis. Methods: A cohort of 500 patients (mean age: 63.9 ± 11.2 years) diagnosed with osteoarthritis and having a medical prescription for dual-energy X-ray absorptiometry (DXA) were recruited for the study. All patients underwent BMD measurements at lumbar spine and femoral sites by both DXA and REMS techniques. Results: The T-score values for BMD at the lumbar spine (BMD-LS) by DXA were significantly higher with respect to BMD-LS by REMS across all OA severity scores, and the differences were more pronounced in patients with a higher degree of OA severity (*p* < 0.001). Furthermore, the percentage of subjects classified as “osteoporotic”, on the basis of BMD by REMS was markedly higher than those classified by DXA, both when considering all skeletal sites (39.4% vs. 15.1%, respectively) and the lumbar spine alone (30.5% vs. 6.0%, respectively). A similar pattern was observed when OA patients were grouped according to the Kellgren–Lawrence grading score. Conclusions: The findings from our study indicate that, in a population with varying severity levels of osteoarthritis, REMS demonstrated a higher capability to diagnose osteoporosis compared to DXA, and this could lead to earlier intervention and improved outcomes for patients with bone fragility, reducing the likelihood of fractures and associated complications.

## 1. Introduction

Osteoarthritis (OA) and osteoporosis (OP) are two very common age-related diseases that present a significant socioeconomic problem due to the reduced quality of life of patients and high rates of disability [1]. The relationship between OA and OP, despite having been the subject of numerous studies for over three decades, remains controversial today. In fact, many observational and longitudinal studies have shown that there is an inverse relationship between the two diseases, with one disease exerting a protective effect on the other [1,2]. On the contrary, several more recent studies have reported that osteoarthritis is associated with a reduction in bone mineral density (BMD) and an increased risk of fractures [3,4]. Furthermore, a recent meta-analysis by Kim et al. reported that osteoarthritis does not confer a protective effect against osteoporosis, nor does it increase the risk of osteoporosis and fragility fractures compared to a control population of the same age and sex [5]. The measurement of bone mineral density using dual-energy X-ray absorptiometry (DXA) is presently regarded as the gold standard for screening and monitoring bone status. The lumbar region, which is usually used for BMD measurements (L1–L4), is the most reproducible. However, it has been known for many years that the measurement of BMD by DXA at the lumbar spine can give erroneous results due to the presence of degenerative changes from osteoarthritis, especially in adult and elderly patients [6,7]. In particular, Masud et al. reported that the presence of even mild osteophytosis produced an average 24% increase in lumbar spine bone mineral density [6]. In fact, in the context of the spine, osteoarthritis manifestations include disc space narrowing, facet joint arthropathy, end-plate osteophytosis, and vertebral plate sclerosis that can create challenges for accurate BMD measurements by DXA [8,9,10]. To address this issue, there has been a suggestion to measure lumbar spine BMD from the lateral view of DXA, thereby avoiding the projection of inter-apophyseal joints. However, the precision of the lateral scan is compromised by difficulties in patient positioning and frequent overlapping of the last ribs and the iliac crest, making it not recommended as a measurement site for diagnosing osteoporosis [11]. Moreover, the exclusive use of femoral scans can sometimes present limits in terms of lower diagnostic sensitivity and the difficulty of positioning elderly patients or those with motor disorders [6]. A further possibility could be represented by high-resolution computerized tomography, but this technique due to the high costs and the high radiation to the patient does not appear feasible for routine clinical use [12]. Some authors have reported that the trabecular bone score (TBS), a parameter obtained through the variations of the pixel gray levels in the DXA images and correlated to some aspects of the vertebral microarchitecture, unlike the lumbar spine BMD, is poorly influenced by osteoarthritis and could be a better predictor of fractures than spine BMD [13,14].

Currently, the method that has the greatest potential to overcome the problem of overestimation of lumbar BMD by DXA in patients with spinal osteoarthritis is Radiofrequency Echographic Multi Spectrometry (REMS). REMS is a non-ionizing and relatively recent technology for bone assessment based on the analysis of native raw, unfiltered ultrasound signals obtained during an echographic scan of the lumbar vertebrae and proximal femur. Examining native, unfiltered ultrasound signals allows for the comprehensive acquisition of information regarding the properties of the assessed tissues. Through the comparison of spectral characteristics in the analyzed signals with established reference spectral models for both pathological and normal conditions, it becomes possible to ascertain bone density [15,16]. A recent study by Messina et al. [17], using data from two consecutive REMS acquisitions by the same operator or two different operators, respectively, demonstrated that both precision (expressed as the root-mean-square coefficient of variation) and repeatability (expressed as the least significant change) of REMS were not inferior to those of DXA. Furthermore, REMS, since it does not use ionizing radiation, can be employed even in subsets of patients for which DXA is not recommended, such as pregnant women, breast-feeding women, and children. In fact, a recent longitudinal study carried out in pregnant women reported a significant reduction in BMD measured by REMS technology between the first and third trimesters of pregnancy [18]. Several cross-sectional and longitudinal studies have demonstrated that the precision and diagnostic accuracy of REMS are not inferior to those of DXA [19,20]. In particular, a large Italian multicenter study reported that REMS had a slightly higher sensitivity and specificity than DXA in identifying women subject to fragility fractures over a 5-year follow-up period [19].

Moreover, a few preliminary reports have documented that REMS technology is able to improve the accuracy of the diagnosis of osteoporosis in cases where lumbar BMD by DXA is impaired by the presence of artifacts from vertebral fractures or osteoarthritis [21,22,23]. Therefore, on the basis of these preliminary data, we conducted the DEMETER study.

### Aim of the Study

The DEMETER study (evaluation of bone mineral DEnsity by radiofrequency echographic Multi spEctrometry technology in osTEoaRthritis) aims to evaluate the usefulness of REMS technology in patients with osteoarthritis. In particular, the aim of the present study was twofold: firstly, to determine the influence of the degree of severity of osteoarthritis on BMD values measured with both the DXA technique and the REMS technique; secondly, to assess whether the use of the REMS technique may improve the identification of osteoporosis in subjects with osteoarthritis.

## 2. Materials and Methods

### 2.1. Study Design and Population

For this study, we enrolled patients with osteoarthritis of both sexes and aged between 45 and 80 years who had a medical prescription for a spinal and/or femoral DXA investigation and had been referred to the Outpatients Clinics of Internal Medicine Unit and Rheumatology Unit at the University Hospital of Siena (Italy). In the time period between January 2021 and June 2023, a cohort of 500 consecutive patients (349 women and 82 men, mean age: 63.9 ± 11.2 years) were enrolled. The study was conducted in accordance with the 1964 Helsinki Declaration, and its later amendments and research protocol received approval from the Ethics Committee of Siena University Hospital (ID-21211/21). Informed written consent was obtained from all participants. Prior to statistical analysis, all collected data underwent anonymization procedures.

The inclusion criteria comprised individuals who met the following conditions: age falling within the range of 45 to 80 years, postmenopausal status, a body mass index (BMI) less than 40 kg/m^2^, and confirmation of osteoarthritis at the lumbar spine through radiological examination. Exclusion criteria encompassed a history of prior treatment with anti-osteoporosis drugs (excluding calcium and vitamin D supplements), the presence of diseases with skeletal involvement (such as cancer, multiple myeloma, hyperparathyroidism), or undergoing therapies that could impact bone metabolism (such as glitazones, glucocorticoids, anticonvulsants, etc.). Out of the initial 500 enrolled patients, 36 were excluded due to strict inclusion/exclusion criteria or their decision to decline participation. Additionally, 33 subjects were excluded from the analysis due to the inadequate quality of REMS or DXA measurements. Consequently, a total of 431 patients with radiological osteoarthritis of the lumbar spine were included in the subsequent analysis.

Patients with comorbidities and infirmities capable of preventing normal daily motor autonomy were also excluded. Throughout the study, height and weight were systematically measured using standardized procedures. The BMI was then computed by dividing the weight in kilograms by the square of the height in meters.

### 2.2. Digital X-ray Radiography

The retrospective evaluation of osteoarthritis of the lumbar spine was independently performed by two of the authors (C.C. and E.C.) with specific expertise. When there were differing opinions during the retrospective evaluation of osteoarthritis of the lumbar spine, a consensus was achieved through discussion with a radiologist. The authors interpreted the results using the Kellgren and Lawrence grading scale (K-L) for OA without assessing joint space narrowing and osteophytes separately [24]. K-L grades range from 0 to 4. They represent absence/doubtful = 0/1, minimal = 2, moderate = 3, and severe = 4 osteoarthritis, respectively. In our study, osteoarthritis was considered to be present when the K-L grade was at least 2 [24]. The flow chart showing the distribution of participants in the DEMETER study is presented in Figure 1.

### 2.3. Dual-Energy X-ray Absorptiometry

In all participants, we carried out BMD measurements at the lumbar spine (LS-BMD), femoral neck (FN-BMD), and total hip (TH-MD) using a dual-energy X-ray absorptiometry device (Discovery W, Hologic, Waltham, MA, USA). The medical reports incorporated the BMD value derived from DXA (expressed in g/cm^2^) along with the corresponding T-score value determined using the reference database for Caucasian Italian women [25]. All DXA acquisitions were performed in accordance with the recommendations of the guidelines and the manufacturer’s technical indications. Osteoporosis and osteopenia were diagnosed in accordance with the World Health Organization’s (WHO) definition: a T-score below −2.5 indicated osteoporosis, and a T-score between −1.0 and −2.5 denoted osteopenia.

### 2.4. Radiofrequency Echographic Multispectrometry (REMS)

Bone mineral density assessments through REMS scans were carried out using a specialized echographic device (EchoStation, Echolight Spa, Lecce, Italy), featuring a convex transducer with a nominal frequency of 3.5 MHz. Before carrying out an assessment of the bone status with REMS technology, the operator must place the probe on the abdomen or hip to visualize the vertebrae or the proximal femur, respectively, and be able to set the depth of the scan and the focus of the probe in accordance with the patient’s physical constitution. For the assessment of lumbar spine BMD, the operator first positions the probe under the sternum to visualize the first lumbar vertebra (L1) and then orients the probe downwards to L4, following the instructions provided on the screen by the software of the device. The lumbar scan requires approximately 80 s (20 s per vertebra) and is necessarily followed by data processing, which requires a further 1–2 min. For the femoral scan, the probe is positioned parallel to the head–neck axis of the femur to visualize the typical profile of the proximal femur. The femoral scan requires approximately 40 s and is necessarily followed by data processing, which requires a further 1–2 min. Following this, the software identifies the sought-after bone interfaces within the acquired frames and delineates regions of interest for diagnostic evaluation. The analysis of spectra from individual scan lines enables the automatic exclusion of signals associated with artifacts, such as calcifications or osteophytes, thanks to the recognition of unexpected spectral features. The selected measured data are ultimately condensed into a patient-specific spectrum of the designated bone target. This spectrum undergoes an advanced comparison with reference spectral models extracted from a dedicated database, ensuring matching criteria for gender, age, skeletal site, and BMI. Indeed, the comparison procedure identifies the spectral modifications introduced by the physical properties of the bone structure, as reflected in the backscattered ultrasound signals. This process leads to the estimation of BMD and subsequently allows for diagnostic classification as healthy, osteopenic, or osteoporotic [16,17]. The intra-operator precision of REMS was 0.47% at the spine and 0.32% at the proximal femur evaluation, whereas the inter-operator precision was 0.55% and 0.51% at the lumbar spine and proximal femur, respectively [20]. Moreover, all REMS measurements were carried out by two of the authors (MDTP and AA), experts in this field.

### 2.5. Statistical Analysis

The values in the study are presented as the mean ± standard deviation (SD). The normality of the distribution of outcome variables was assessed using the Kolmogorov–Smirnov test. Clinical data and initial values of the measured variables in the study groups were compared using Student’s *t*-test and Mann-Whitney U-test, depending on the appropriateness of the data distribution. Categorical variables were subjected to comparison using the Chi-square test or Fisher’s exact test, as deemed appropriate. Associations between different parameters were examined through Pearson’s correlation or Spearman’s correlation, as appropriate, or via partial correlation analysis. All statistical analyses were performed using the SPSS statistical package for Windows version 16.0 (SPSS Inc., Chicago, IL, USA).

## 3. Results

The demographic and clinical characteristics of the study population are shown in Table 1. As expected, subjects with a more severe OA score (K-L score = 3 and K-L score = 4) were older and presented BMD values at all skeletal sites higher with respect to those with mild osteoarthritis (K-L score = 2) or absence/doubtful (K-L = 0/1). Instead, BMD values obtained from the REMS technique showed an opposite trend compared to those obtained from the DXA technique. Finally, BMI presented a slight and non-significant increase in patients with more advanced OA.

Figure 2 illustrates the BMI-adjusted partial correlation between BMD-LS T-scores obtained via DXA and REMS techniques in relation to osteoarthritis scores. There is a noticeable high correlation between the T-score values derived from both techniques at the lumbar spine across different osteoarthritis scores. (K-L score = 0/1, r = 0.98, *p* < 0.0001; K/L score = 2, r = 0.85, *p* < 0.001; K/L score = 3, r = 0.98, *p* < 0.0001; and K/L score = 3, r = 0.45, *p* < 0.01, respectively). As expected, less severe OA scores had better significance.

The values of T-score BMD at the lumbar spine, measured by the DXA and REMS technique, according to the OA score, are shown in Figure 3. The values of the T-score BMD-LS by DXA were significantly higher with respect to BMD-LS by REMS at all OA scores; in particular, the difference became more significant as the degree of severity of OA increased (*p* < 0.001).

Figure 4 illustrates osteoarthritis patients categorized as “osteoporotic”, “osteopenic”, or “normal” based on BMD T-score values obtained through both DXA and REMS techniques, either at the lumbar spine alone (Figure 4A) or across all skeletal sites (Figure 4B). Notably, the REMS technique allows a larger percentage of OA patients to be classified as osteoporotic or osteopenic at the lumbar spine compared to DXA (30.5% and 52.0% vs. 6.0% and 34.5%, respectively). Conversely, the percentage of subjects classified as normal by DXA was higher than that by REMS (59.5% versus 17.5%, respectively) (Figure 4A). Similarly, when considering all skeletal sites, the percentage of patients with OA classified as osteoporotic was significantly higher with REMS compared to DXA (39.4% vs. 15.1%, respectively). On the contrary, the percentage of subjects with OA classified as osteopenic or normal at all skeletal sites by DXA was similar and higher than that by REMS (51.1% and 30.4% vs. 54.5% and 9.5%, respectively) (Figure 4B).

Figure 5 shows the percentage of patients classified as having osteoporosis, osteopenia, or normal on the bases of BMD at the lumbar spine (Figure 5A) or at all skeletal sites (Figure 5B) according to the K-L grading score. A similar trend was observed when the patients were grouped according to the severity of the OA grade.

## 4. Discussion

The most important finding of this study is represented by the fact that LS-BMD by REMS is not affected by the presence of lumbar spine osteoarthritis, even to a moderate and severe degree. Instead, as expected, the LS-BMD by DXA increases significantly with the Kellgren and Lawrence grade.

This study also demonstrated that hip BMD by DXA increases with the severity of lumbar spine osteoarthritis. This finding appears in agreement with both a French study that observed that in postmenopausal women, increasing the severity of lumbar osteoarthritis was associated with a significant increase in femoral BMD [13] and a previous study carried out in the general population reporting that patients with osteoarthritis at the lumbar spine showed a higher femoral neck BMD [1]. Over the years, several hypotheses have been advanced to explain this relationship, such as the greater bone size or higher BMI in patients with osteoarthritis resulting in subchondral bone osteosclerosis and consequently overestimation of BMD due to measurement artifacts. However, several studies have reported that the increase in femoral BMD by DXA observed in individuals with high-grade osteoarthritis at the lumbar spine was independent of BMI values and obesity [1,13]. Indeed, these factors do not appear to have an effect on femoral BMD measurements by the REMS technique. In fact, in our study, the values of BMD by REMS at the femoral sites showed a different trend. In particular, BMD by REMS at both the femoral neck and total hip tended to decrease with increasing osteoarthritis grade. While a clear explanation for the observed pattern of femoral BMD by REMS remains elusive, it is noteworthy that a similar trend has also been observed in studies involving trabecular bone score. In fact, some studies have reported that TBS was lower in patients with moderate or severe OA than in those with mild OA [13]. Based on these observations, it is possible to hypothesize that both TBS and REMS could capture the worsening of the structural and qualitative characteristics of the bone tissue in patients with severe osteoarthritis [4]. Furthermore, this study highlights that REMS technology, through the analysis of raw ultrasound unfiltered signals, can effectively identify and automatically eliminate artifacts arising from osteoarthritis, such as osteophytes, facet-joint issues, and end-plate sclerosis, particularly affecting the lumbar spine and especially the more caudal vertebrae [6,8,9,26]. These findings confirm REMS technology’s capability not only to address artifacts related to osteoarthritis but also those associated with spondylitis, vertebral fractures, and vertebroplasty or kyphoplasty operations. This capability enhances the accuracy of osteoporosis diagnosis and, consequently, a better definition of fragility fracture risk [21,22]. However, it’s important to note that the absence of published data on the accuracy of REMS in individuals with a BMI greater than 40 kg/m^2^ may represent a limitation in the use of REMS in severely obese patients.

Another important finding of this study is the ability of REMS technology to diagnose an osteoporosis condition in a much higher number of patients with osteoarthritis compared to DXA. Until recently, the vast majority of studies have focused on the association between osteoarthritis of either hip or knee and BMD, supporting the hypothesis that OA and OP may have a protective effect of one against the other. On the other hand, literature data on the relationship between the severity of OA at the lumbar spine and femoral BMD are very scarce. A long-term study carried out on 5000 elderly women found no difference in BMD values at the axial and appendicular sites between subjects with or without hand osteoarthritis; furthermore, this study showed an increase in vertebral and femoral fractures in women with hip osteoarthritis, which was attributed to the increase in falls [27]. Several studies reported that an increase in BMD did not translate into a reduction in fracture risk [28,29]. In particular, in the Women’s Health Initiative (WHI) study, the risk of any fracture and spine fracture were increased by 1.1-fold and 1.2-fold, respectively, in subjects with self-reported OA [28]. On the other hand, it is important to underline how the alterations induced by spinal osteoarthritis (osteophytes, facet sclerosis) can shield osteoporosis, consequently leading to misinterpretation of BMD [4]. It is also possible to hypothesize that the modifications in geometry and microarchitecture induced by osteoarthritis may interfere with bone quality and strength [4,30]. The ability of REMS to diagnose osteoporosis in more patients with osteoarthritis than DXA may allow better identification of patients at increased risk of fracture requiring anti-osteoporotic drug therapy. Other strengths of REMS technology compared to DXA, which still remains the gold standard for the diagnosis of osteoporosis, are represented by the absence of ionizing radiation, portability, and lower cost.

This study has some limitations that should be considered. Firstly, the cross-sectional design of the study precludes the establishment of causal relationships between the measured parameters. Secondly, the study was conducted exclusively in Caucasian patients with a BMI ranging from 18.5 to 39.9 kg/m^2^, thereby limiting the generalizability of our findings to individuals with a BMI equal to or greater than 40 kg/m² or those of non-Caucasian descent. Thirdly, the absence of a third reference technique, like quantitative computed tomography (QCT), could provide additional assurance and help validate the accuracy of BMD measurements obtained through REMS. Despite these limitations, the study boasts several strengths. Firstly, it was conducted by a single center, encompassing a large cohort of patients with osteoarthritis. Secondly, the Kellgren and Lawrence grade was centrally evaluated by two readers. Thirdly, both DXA and REMS scans were performed by two experienced operators, ensuring consistency and reliability in the data collection process.

## 5. Conclusions

This is the first large study on the clinical utility of REMS technology in assessing the bone status of patients with osteoarthritis, grouped on the basis of Kellgren and Lawrence grade. In our patients with osteoarthritis, REMS has been shown to be able to diagnose osteoporosis in significantly more cases than DXA, and this could improve the possibility of treating patients with bone fragility that commonly escapes treatment. Additional studies are needed to validate these initial findings and ascertain the role of REMS technology in the assessment of osteoporosis and fragility fracture risk in patients with osteoarthritis.

## Figures and Tables

**Figure 1 diagnostics-14-00523-f001:**
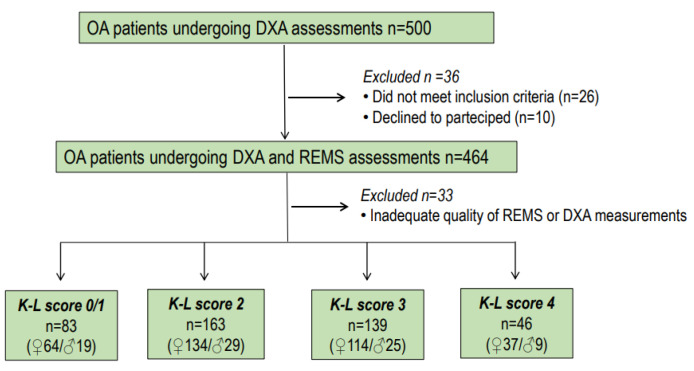
Flow chart showing the distribution of participants to the DEMETER Study.

**Figure 2 diagnostics-14-00523-f002:**
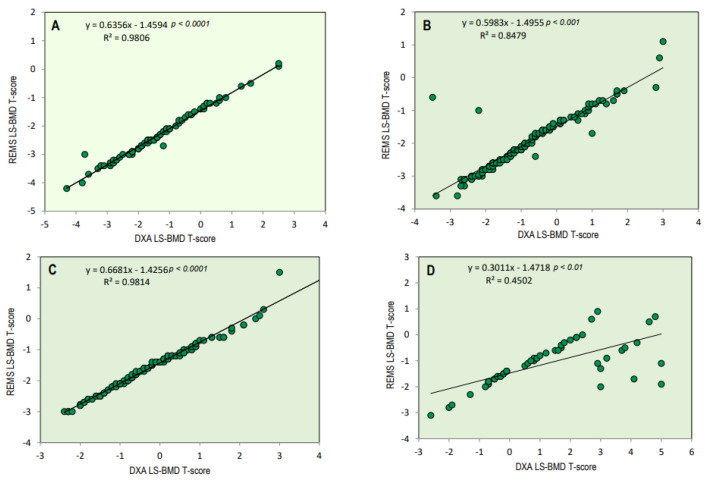
BMI-adjusted partial correlation between BMD-LS T-score by DXA technique and BMD-LS T-score by REMS technique according to osteoarthritis score (K-L score 0/1 = (**A**), K-L score 2 = (**B**), K-L score 3 = (**C**), and K-L score 4 = (**D**)).

**Figure 3 diagnostics-14-00523-f003:**
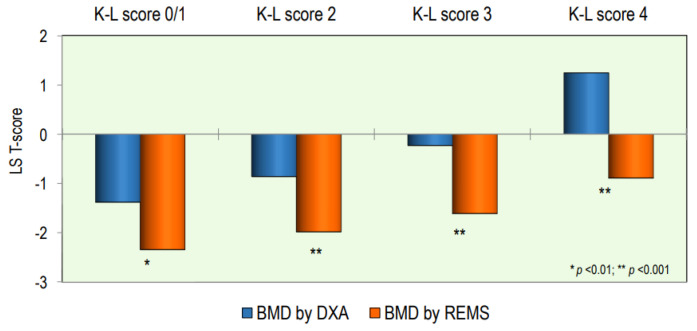
Values of BMD expressed as a T-score at the lumbar spine by DXA and REMS techniques grouped by OA score.

**Figure 4 diagnostics-14-00523-f004:**
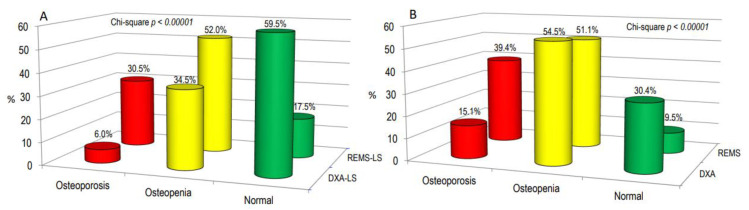
The proportion of osteoarthritis (OA) patients categorized as “osteoporotic”, “osteopenic”, or “normal” based on BMD T-score values acquired through DXA and REMS techniques specifically at the lumbar spine (**A**) or across all skeletal sites (**B**).

**Figure 5 diagnostics-14-00523-f005:**
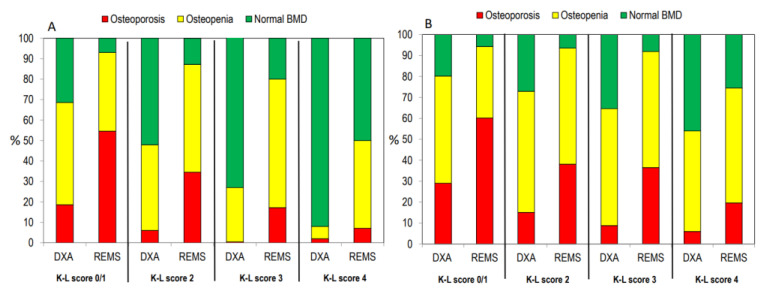
Percentage of patients classified as “osteoporotic”, “osteopenic”, or “normal” at the lumbar spine alone (**A**) or at all skeletal sites (**B**) according to the K-L score.

**Table 1 diagnostics-14-00523-t001:** Anthropometric, clinical, and densitometric characteristics of the study population grouped according to the Kellgren–Lawrence score.

	*All*	*K-L* *Score 0/1*	*K-L* *Score 2*	*K-L* *Score 3*	*K-L* *Score 4*	*p*
Age (yrs)	63.9 ± 11.2	59.2 ± 11.4	62.8 ± 11.0	65.9 ± 10.5	70.5 ± 9.4	0.0001
Weight (Kg)	70.3 ± 14.9	73.3 ± 17.4	68.8 ± 14.3	69.5 ± 14.1	72.3 ± 14.2	0.086
Height (cm)	163.6 ± 8.6	166.1 ± 9.4	163.0 ± 8.2	162.7 ± 8.1	163.6 ± 9.7	0.025
BMI (Kg/m^2^)	26.1 ± 4.6	26.3 ± 4.9	25.8 ± 4.4	26.1 ± 4.6	27.0 ± 4.3	0.379
DXA LS-BMD (g/cm^2^)	1.048 ± 0.198	0.956 ± 0.169	1.002 ± 0.153	1.078 ± 0.166	1.269 ± 0.269	0.0001
DXA FN-BMD (g/cm^2^)	0.782 ± 0.157	0.770 ± 0.155	0.761 ± 0.145	0.793 ± 0.158	0.841 ± 0.182	0.015
DXA TH-BMD (g/cm^2^)	0.900 ± 0.157	0.870 ± 0.146	0.888 ± 0.151	0.913 ± 0.159	0.952 ± 0.176	0.018
REMS LS-BMD (g/cm^2^)	0.854 ± 0.108	0.892 ± 0.127	0.858 ± 0.103	0.832 ± 0.095	0.835 ± 0.100	0.0001
REMS FN-BMD (g/cm^2^)	0.673 ± 0.126	0.717 ± 0.147	0.677 ± 0.122	0.647 ± 0.116	0.660 ± 0.113	0.001
REMS TH-BMD (g/cm^2^)	0.809 ± 0.145	0.858 ± 0.163	0.812 ± 0.138	0.781 ± 0.138	0.794 ± 0.132	0.002

## Data Availability

The data that support the findings of this study are available from the corresponding author (C.C.) for privacy.

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
