# Peer review of "The Advantages of Radiofrequency Echographic MultiSpectrometry in the Evaluation of Bone Mineral Density in a Population with Osteoarthritis at the Lumbar Spine"

_diagnostics, 2024, doi:10.3390/diagnostics14050523_

Round 1
Reviewer 1 Report
Comments and Suggestions for Authors
Interesting article on a little covered topic.
The topic of the manuscript is evaluation of the use of REMS in the evaluation of BMD in a population with osteoarthritis at the lumbar spine. There are few related articles. My comments: Punctuation and style should be improved throughout the text. Abstract: Well written. Introduction: well written, extensive literature review Please ad research objectives and hypothesis. Materials and methods: clearly and extensively presented. Large study group. Good description of the research methodology. Well-selected statistical tools. Clear and good quality figures. Results: Well written. Clear, legible and good quality tables and figures. The discussions: well written, extensive literature review. Work limitations clearly shown. Conclusions: Well and clearly written. The references are appropriate.Comments on the Quality of English Language Punctuation and style should be improved throughout the text.
Author Response
Response to Referee 1
- First of all we want to thank the referee for his comments and suggestions
- Punctuation and style have been improved throughout the text.
- The research objectives in the introduction section have been more clearly reported in a separate paragraph

Reviewer 2 Report
Comments and Suggestions for Authors
Dear Editor,
Thank you for inviting me to review the manuscript entitled: “The advantages of Radiofrequency Echographic MultiSpectrometry (REMS) in the evaluation of BMD in a population with osteoarthritis at the lumbar spine” submitted for publication in Diagnostics.
The authors aimed in this context to evaluate the use of radiofrequency echographic multi spectrometry as a tool for enhancing the diagnosis of osteoporosis in patient with osteoarthritis in the lumbar spine.
Some comments below:
First of all, regarding the title- the authors should not use abbreviations “(REMS)”, as well as “BMD”.
The abstract is well-written
In the introduction, the authors make a good brief review of the current literature, providing the basic knowledge for the study. My only comment would be, at the end, to make clearer the aim of the study. Use a separate small paragraph and expand this a bit.
In the Materials and Methods section:
Line 93, rephrase to “..who had a medical..”
You should rephrase lines 91-96, it is unclear whether the goal was to enroll 500 consecutive patients during the study period or you enrolled all patients with these characteristics from 2021 to 2023?
In lines 99-102 the authors describe the inclusion criteria. It is stated there that “postmenopausal status” is one of them. The study refers only to females? If so this should be stated earlier on? Why not include men with OA in the lumbar spine as well?
Section 2.2 is very useful, explaining a crucial aspect of the study’s methods- Good job!
Line 174, a “ is missing.
In general, the study is interesting and well-written. My main comments have to do with the presentation of the methodology.
With best regards
Author Response
- According to the suggestion of the referee we remove the abbreviations by the title
- According to the suggestion of the referee we have better reported the aim of the study in a separate paragraph at the end of the Introduction.
- Materials and Methods: The paragraph that includes lines 91 to 96 has been rewritten more clearly.
- First of all we want to thank the referee because his comment allowed us to better define the characteristics of the study population which included both sexes even if the vast majority were women.

Reviewer 3 Report
Comments and Suggestions for Authors
Thank you for the opportunity for reviewing this paper.
This study is well designed and very informative study.
However, I have one question about this paper.
T-score of BMD measured using DXA is well known and widely used in clinical practice.
However, is there validaty for the T-score of BMD measured using REMS?
Is it possible to prescribe medicaiton for osteoporosis using the T-score of BMD measured using REMS?
Author Response
A good agreement between the T-score values of BMD by DXA and the T-score values of BMD by REMS was reported several years ago in an Italian study by Di Paola Fet al.
These results were confirmed by Cortet B et al. in a wider European population reporting that areas under the curve (AUCs) of the Receiver Operating Characteristic (ROC) curve evaluating the ability to discriminate groups of patients with previous osteoporotic fracture using DXA T-score and REMS T-score values were higher for REMS at both the femur and lumbar spine.
A recent prospective observational study by Adami G et al. carried out on 1370 women followed for 5 years confirmed that the T-scores obtained by REMS were an effective predictor of fragility fractures, – Moreover, Bland-Altman plots comparing REMS and DXA T-score measurements for both lumbar spine and femur have the diagnostic agreement.
In Italy REMS has been acknowledged in the Guidelines by the National Institute of Health as a valid diagnostic alternative for the management of osteoporosis diagnosis and risk stratification, but so far it has not yet been recognized for the reimbursement of drugs.

Round 2
Reviewer 2 Report
Comments and Suggestions for Authors
Dear Editor,
The authors have made the requested edits.
The manuscript looks fine.
With best regards